# Predicting PM$_{2.5}$ in the Northeast China Heavy Industrial Zone: A Semi-Supervised Learning with Spatiotemporal Features

**Hongxun Jiang** **, Xiaotong Wang and Caihong Sun \***

School of Information, Renmin University of China, Beijing 100872, China
* Correspondence: chsun@ruc.edu.cn; Tel.: +86-10-82500904

**Abstract:** Particulate matter PM$_{2.5}$ pollution affects the Chinese population, particularly in cities such as Shenyang in northeastern China, which occupies a number of traditional heavy industries. This paper proposes a semi-supervised learning model used for predicting PM$_{2.5}$ concentrations. The model incorporates rich data from the real world, including 11 air quality monitoring stations in Shenyang and nearby cities. There are three types of data: air monitoring, meteorological data, and spatiotemporal information (such as the spatiotemporal effects of PM$_{2.5}$ emissions and diffusion across different geographical regions). The model consists of two classifiers: genetic programming (GP) to forecast PM$_{2.5}$ concentrations and support vector classification (SVC) to predict trends. The experimental results show that the proposed model performs better than baseline models in accuracy, including 3% to 18% over a classic multivariate linear regression (MLR), 1% to 11% over a multi-layer perceptron neural network (MLP-ANN), and 21% to 68% over a support vector regression (SVR). Furthermore, the proposed GP approach provides an intuitive contribution analysis of factors for PM$_{2.5}$ concentrations. The data of backtracking points adjacent to other monitoring stations are critical in forecasting shorter time intervals (1 h). Wind speeds are more important in longer intervals (6 and 24 h).

**Keywords:** data mining; genetic programming; machine learning; support vector classification

## 1. Introduction

A major concern among the Chinese public has been increasing air pollution as a result of rapid industrialization and urbanization. Among the pollutants, particulate matter (PM)$_{2.5}$, which refers to particles less than 2.5 μm in diameter, has become the biggest threat to human health. The prediction of PM$_{2.5}$ concentrations has attracted both scientists and society. The study in this field starts at the end of the 20th century and has reached a new height for machine learning exploding in recent years. The most widely used models are multivariate linear regression (MLR), support vector regression (SVR), and artificial neural networks (ANNs). Despite so many approaches and models, most are still far from good enough in accuracy and robustness.

This paper collects real-world data to train machine learning models, including hourly PM$_{2.5}$ concentration from January to March 2022, from all air-monitoring sites in Shenyang, China., Predictions are based on meteorological data, especially daily sunlight, which has an essential influence on forming PM$_{2.5}$ concentrations [1,2]. Historical PM$_{2.5}$ concentrations from the target site and nearby sites, such as backtracking points, are critical aspects of the proposed models. We separate the task of predicting PM$_{2.5}$ into two models: one for exploring the PM$_{2.5}$ concentrations and the other for forecasting the trend. Our model predicts concentrations mainly using genetic programming (GP) after using the shortest-distance clustering method to cluster the correlation coefficients of PM$_{2.5}$ concentrations at each site. In addition, we provide support vector classification (SVC) to forecast the trend, i.e., an increase or a decrease in the PM$_{2.5}$ index, in advance of time intervals of 1 h, 6 h, and 24 h.

The remainder of the paper is organized as follows. Section 2 briefly presents a literature review of machine learning models. Section 3 proposes a semi-supervised learning model to predict PM$_{2.5}$, with two classifiers: a GP for concentrations and SVC for trends. In Section 4, we conduct experiments to verify our proposed models. Finally, some concluding remarks and contributing insights are given in Section 5.

## 2. Related Works

The first attempt to predict air quality can be traced back to Miller et al. (1976) [3], who used two models based on Gaussian dispersion equations to indicate carbon monoxide concentrations. Recently, air quality prediction has three mainstreams: "observation" in meteorological and atmospheric sciences [4–6], which applies satellite observations and automatic forecasting systems just like a weather forecast; "formation" in chemistry [7], which focuses on the procedures of pollutants that form and predict the pollutant concentrations based on chemical reactions; and "derivation" in computer science and engineering, which applies machine learning through historical data. The models widely used are neural networks, support vector machines, and others, as described in the following.

The neural network (NN) model is a kind of learning with similar behavior characteristics to those of a biological neural network. Grivas et al. (2006) predicted hourly PM$_{10}$ concentrations in Athens, wherein NN models outperformed the MLR model [8]. Comrie (2012) compared NN models with an MLR model to predict ozone concentrations and concluded that an NN was slightly better [9]. McKendry (2011) reported on applications indicating PM$_{10}$, PM$_{2.5}$, and ozone levels using a NN [10]. Ordieres et al. (2005) used NN models to predict PM$_{2.5}$ concentrations on the US–Mexico border and verified that NN models were more accurate than a persistence model and MLR [11].

The support vector machine (SVM), proposed by Cortes and Vapnik (1995), is a supervised learning model usually for pattern recognition, classification, and regression analysis [12]. Xu et al. (2014) adopted an SVR-based ensemble learning to predict PM$_{2.5}$ concentrations [13]. They considered the influences of other monitoring sites, and closer sites have higher weights. Zhao et al. (2013) built two classification models by NNs and SVC to predict whether the 24 h average PM$_{2.5}$ concentrations in Hong Kong would be over 40 $\mu g \cdot m^{-3}$ [14].

Furthermore, there are some other methods or concepts used for air quality prediction. Vardoulakis et al. (2003) proposed that "street canyons", or the spatial structures of streets, affect air quality [15]. Dong et al. (2009) presented a methodology based on hidden semi-Markov models to predict high PM$_{2.5}$ concentration values [16]. The concept of genetic programming (GP) is one of the most influential methods of machine learning. Genetic programming mimics biological evolution and is closely related to genetic algorithms. J. R. Koza (1994) introduced the principles and applications of GP [17]; from then on, GP has been applied to many fields. Kishore et al. (2000) explored the feasibility of GP in multi-category pattern classification and found that the performance of the GP classifier was better than that of traditional global learning [18]. Brameier et al. (2001) compared GP with NNs in medical data mining and found that GP performed comparably in both classification and generalization [19]. Muni et al. (2006) proposed a GP algorithm for feature selection and then constructed a classifier [20]. Tay et al. (2008) used GP to solve multi-objective flexible job shop problems; the results showed that the GP-revised rules generally outperformed the single-dispatching rules [21].

GP has also been widely used in predictions or forecasting models. Harris et al. (2003) improved accuracy by using GP for predicting velocities in wetlands and other vegetated areas [22]. Whigham et al. (2001) used GP to model rainfall and runoff and compared GP with the identification of unit hydrographs and component flows from rainfall, evaporation, and stream flow models, which were satisfactory when the rain and runoff were correlated. When there was a poor correlation, GP was more effective [23]. Ong et al. (2005) built credit-scoring models and compared GP with approximate sets, logistic regression, decision trees, and artificial neural networks [24]. They found that GP provided better accuracy and

robustness than the other models. Muttil et al. (2005) presented a real-time model using GP to predict algal blooms, prevent harm to fisheries, and provide a suggestion for the management departments [25]. Baykasoğlu et al. (2008) applied GP to predict limestone's compressive and tensile strengths [26]. They chose a GP rather than a NN because the former produces prediction equations explicitly, while the latter cannot.

## 3. Features Engineering

This paper's data, including $PM_{2.5}$ concentration values, meteorological data, and the latitude–longitude data, came from the official departments in Shenyang and some others from the internet, such as the daily times for sunrise and sunset from http://www.weather.com.cn/ (accessed on 1 January 2022). This section introduces the preprocessing of raw data and selecting features before training models.

### 3.1. Air Quality Data, Meteorological Data, and Their Relevance

According to official air quality standards published on the website of the China National Environmental Monitoring Center in 2012, $PM_{2.5}$ concentration values were divided into six categories: 0 to 35 μg·m$^{-3}$, 36 to 75 μg·m$^{-3}$, 76 to 115 μg·m$^{-3}$, 116 to 150 μg·m$^{-3}$, 151 to 250 μg·m$^{-3}$, and over 251μg·m$^{-3}$, corresponding togood, general, mildly polluted, moderately polluted, severely polluted, and seriously polluted levels, respectively. The percentages of the six categories are shown in Figure 1 (some concentration values greater than 250 μg·m$^{-3}$ are included in the seriously polluted category). The category values are seen as the target of the models.

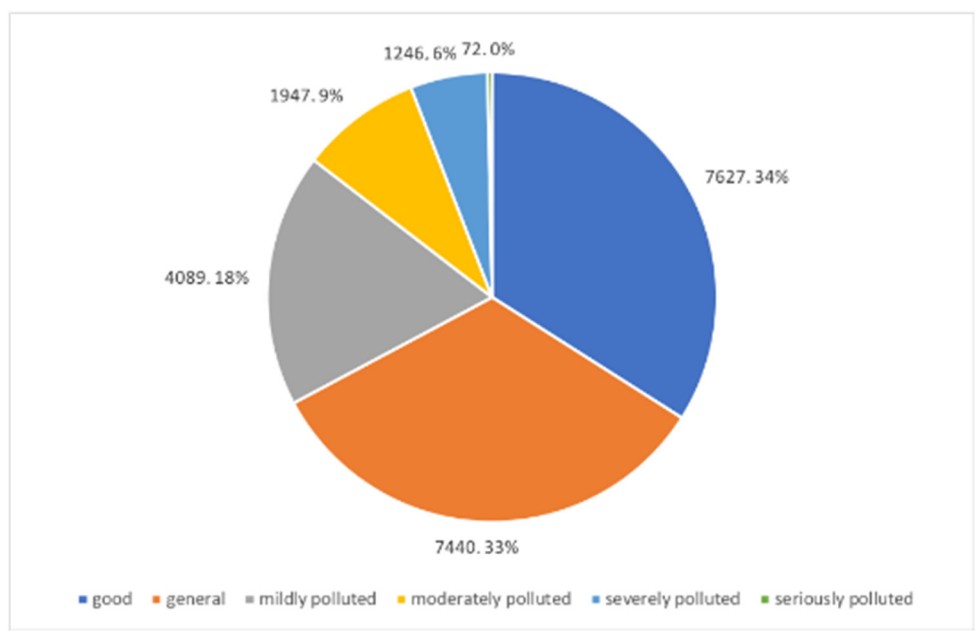

**Figure 1.** The percentages of $PM_{2.5}$ levels in Shenyang.

Meteorological data were gathered at hourly intervals corresponding to the $PM_{2.5}$ concentration values, which were temperature, atmospheric pressure, relative humidity, wind direction, speed. We preliminarily analyzed the correlations of $PM_{2.5}$ concentrations and each meteorological feature using scatter grams shown in Figure 2. Figure 2 indicates that when the temperature was very low (lower than −20 °C), the concentration of $PM_{2.5}$ was also relatively low; when the atmospheric pressure was high, the $PM_{2.5}$ concentrations were relatively low; when the relative humidity was lower than 20%, the $PM_{2.5}$ concentrations were relatively low; and when the wind speed was high, the $PM_{2.5}$ concentrations were relatively low. However, the $PM_{2.5}$ concentrations were not always high when the wind speed was low. The wind direction was measured by an angle, with north at 0°, east at 90°, south at 180°, and west at 270°. The remaining wind directions were calculated from this.

The last picture in Figure 2 also tells us that a correlation between $PM_{2.5}$ concentrations and wind direction cannot be made directly according to the scatter gram.

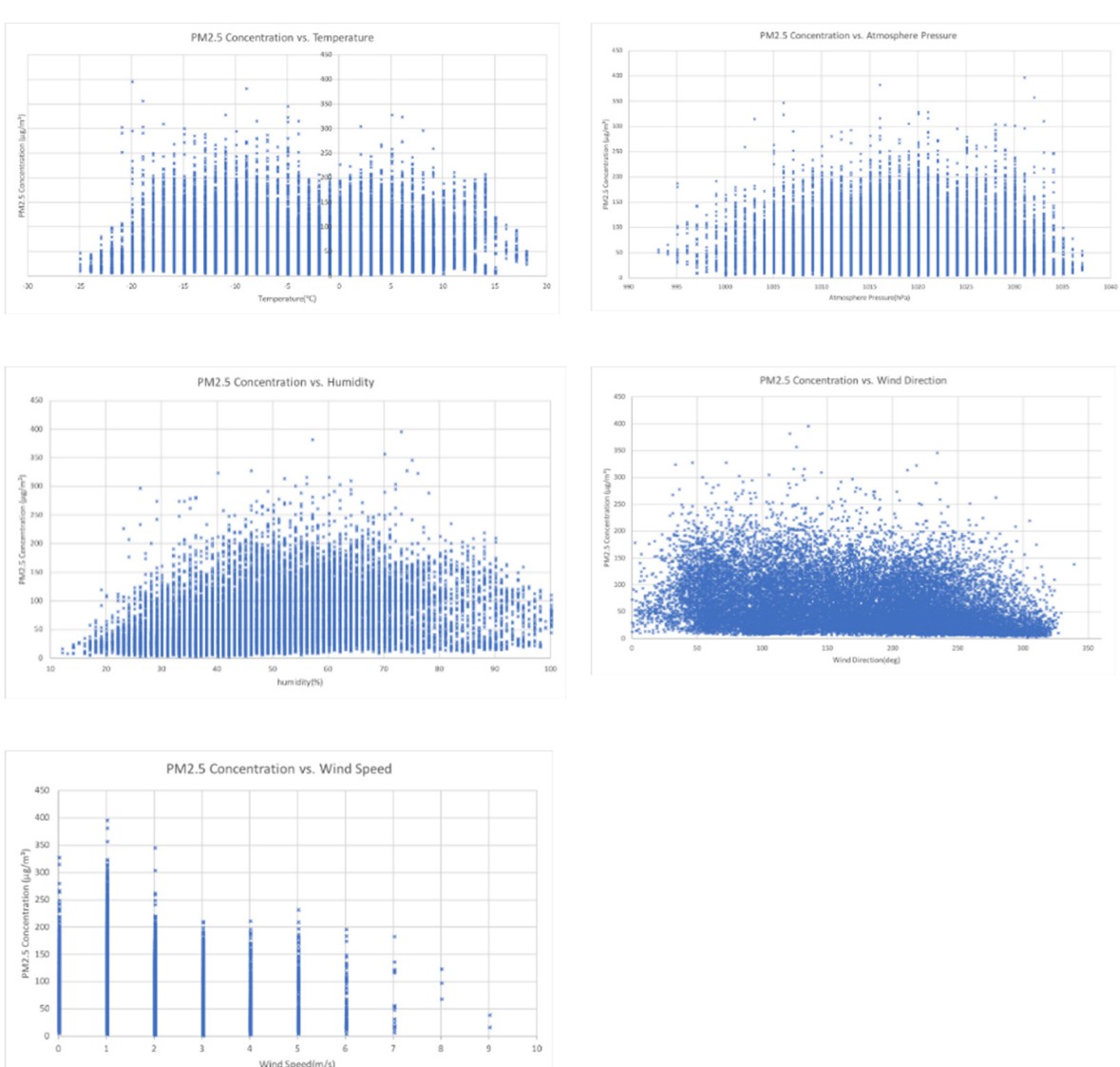

**Figure 2.** The scatter gram of $PM_{2.5}$ vs. other meteorological indices.

### 3.2. Spatiotemporal Data

This paper introduces a spatiotemporal site relevance concept for air quality prediction, based on locations sites, wind direction, and wind speed. Then, $PM_{2.5}$ concentrations of the related sites, called backtracking points, are selected as input features to train the model. For the specifics, see Section 4. For each record of each site, we found the records of the same site 1 h, 6 h, and 24 h earlier. Then, we considered the $PM_{2.5}$ concentration values in the found records as a feature for prediction.

### 3.3. Sunlight Effect

$PM_{2.5}$ concentrations are remarkably affected by sunlight. However, there are many complex effects on light intensity, especially weather. Without the loss of generality, in this study, we simply considered whether the site was in the dark hemisphere. First, we collected data on sunrise and sunset times in Shenyang on a daily basis.. When a record was acquired after sunrise and before sunset, we marked 0 to indicate that the site was in the daylight hemisphere.. Otherwise, if it was later than the time of sunset and earlier than

sunrise on the next day, we recorded 1. Table 1 shows the relationship between the $PM_{2.5}$ index and whether the site was in the dark hemisphere.

**Table 1.** $PM_{2.5}$ concentrations with different hemispheres.

| Hemisphere | Mean Value of $PM_{2.5}$ (μg/m$^3$) | Standard Deviation |
|:---:|:---:|:---:|
| Daylight | 48.3939 | 58.3249 |
| Dark | 45.0552 | 69.3664 |

## 4. The Proposed Model

This paper proposes a semi-supervised learning model for predicting $PM_{2.5}$ concentrations, which consists of two classifiers: genetic programming (GP) to forecast $PM_{2.5}$ concentrations and nu-support vector classification (SVC) to predict trends.

### 4.1. A Genetic Programming Model

A GP model begins with an initial randomly generated population, in which each individual is one predicting model. To achieve a better effect, we specified the basic terminal set and function set for individuals. The terminator set is a function of various parameters and variables, including constants, logic constants, variables, etc. In this research, the terminal set consisted of the data mentioned in Section 3.1 and random numbers. The function set may include basic arithmetic operations, standard programming operations, standard mathematical functions, logic functions, or any other mathematical function. After generating the initial generation, we defined the fitness function, i.e., RMSE in our research, to rank the individuals. Common genetic manipulations, such as replication, crossover, and mutation, were used to generate new offspring populations. Replication involves copying an existing individual directly into a newly created child population without modifying it. Crossover refers to selecting a part of two parents that has a better fitness exchange to produce two new individuals. The process of mutation involves randomly altering a specific part of the parent, thereby creating a new individual.. To finish modeling, we set the maximum number of evolutions and target fitness. When the times of evolution reached the set number or the fitness of the best individual in the population reached the target, the modeling ended, and the prediction model was constructed using the best individual. GP's operating process is shown in Figure 3.

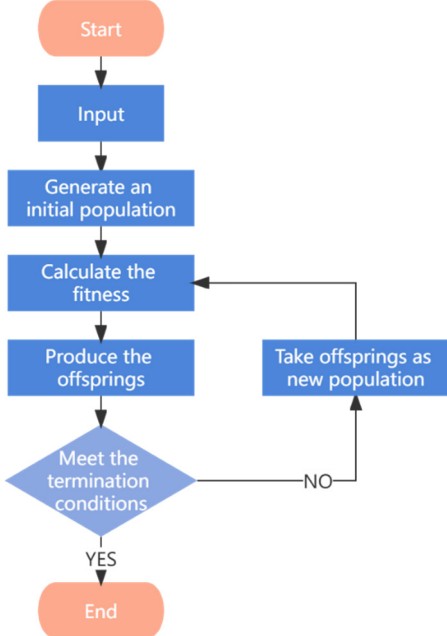

**Figure 3.** The proposed GP flow chart.

Table 2 shows the GP parameters. Some other parameters that were not listed would not likely affect the results; thus, we used the default values. Table 3 shows the meaning of some of the operators in GP.

**Table 2.** The parameter settings of the proposed GP.

| Time Intervals (h) | Operators | Initial Population | Generation |
|:---:|:---:|:---:|:---:|
| 1, 6, 24 | plus, minus, times, sin, cos, sqrt, square, exp, add3, mult3 | 1000 | 500 |

**Table 3.** Examples of GP operators and their meanings.

| Operators | Input Arguments | Output Argument |
|:---:|:---:|:---:|
| exp | a | $e^a$ |
| add3 | a, b, c | $a + b + c$ |
| mult3 | a, b, c | $a \times b \times c$ |

### 4.2. A nu-SVC Model

The SVC model is a type of SVM which can be used for classification purposes. In this paper, we used nu-SVC to predict the changing tendency of $PM_{2.5}$ concentrations. In nu-SVC modelling, we established an optimal decision hyperplane that maximizes the distance between the two types of samples on both sides of the plane, thus providing a good generalization capability for classification problems. For a multidimensional sample set, the system randomly generates a hyperplane and keeps it moving, classifying the samples until the sample points belonging to different classes in the training sample are located exactly on both sides of the hyperplane. There may be many hyperplanes satisfying the condition. SVC formally seeks such a hyperplane while guaranteeing classification accuracy, so that the blank area on both sides of the hyperplane is maximized, thereby achieving an optimal classification of linear separable samples. The parameters of nu-SVC in our research are shown in Table 4.

**Table 4.** The parameter settings of the proposed nu-SVC.

| Time Intervals (h) | SVC Type | Degree | Gamma |
|:---:|:---:|:---:|:---:|
| 1 | nu-SVC | 3 | 0.1429 (without backtracking point) 0.125 (with backtracking point) |
| 6, 24 | nu-SVC | 3 | 0.1429 |

### 4.3. Related Sites and Backtracking Points

For the site-related concept, we used 1 h predictions of $PM_{2.5}$ concentration based on spatial location information and wind speed, which can be calculated as follows. First, we calculated the correlation coefficient matrix of the $PM_{2.5}$ concentrations between monitoring sites. Second, we created a coordinate and put all the sites into it by referring to their latitudes and longitudes, and then calculated the distance between the monitoring. As can be seen from these two appendices, the correlation of the $PM_{2.5}$ concentration between the two sites did not necessarily decrease as the distance between them decreased. Third, we calculated a backtracking point for each record of each site based on a vector whose starting point is the predicted location, whose direction is the opposite of the wind direction, and whose length is the wind speed multiplied by one hour.. Then, we used this vector as a center and drew a fan shape with left and right deviations of 15 degrees. We then selected the sites within the fan shape that have either the highest correlation with the $PM_{2.5}$ concentrations of the predicted site or the shortest distance with the predicted site as the backtracking point. (As for the 6 h and 24 h predictions, because of errors in wind direction and speed, the simplification of wind speed, and the accumulation of these errors

in the 6 and 24 h situations, we abandoned the site-related concept in those two groups of experiments.) Finally, we obtained the PM$_{2.5}$ concentrations of the backtracking point an hour earlier as a basis to predict the PM$_{2.5}$ concentrations of the original site. As Figure 4 shows, we assumed that a north wind blew at site A, and the wind speed was 1 m/s. We then constructed a vector, and B was the backtracking point of site A in this record. So, we considered the PM$_{2.5}$ concentration values of site B as an independent variable in the prediction of the PM$_{2.5}$ concentrations for site A. Furthermore, to decide whether to choose a backtracking point with a correlation coefficient or a distance, we experimented with both ways of selecting a backtracking point and compared the error of the two ways.

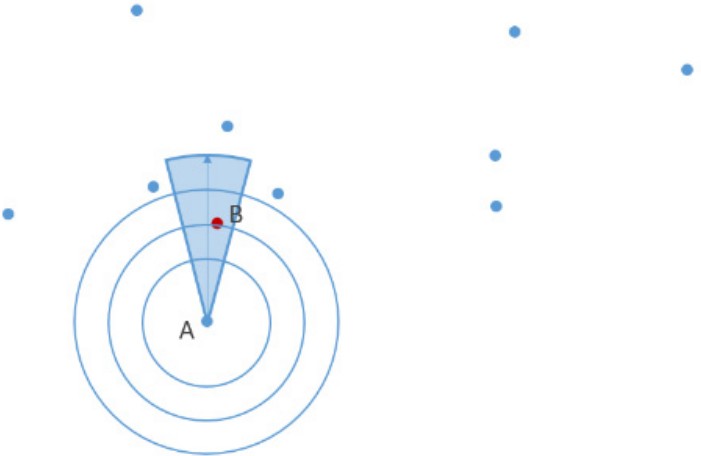

**Figure 4.** An example of the backtracking point B of the target A.

### 5. Experiments and Discussion

From the official departments in Shenyang, we obtained a total of 22,528 datasets across 11 monitoring stations from 01:00 on 1 January to 08:00 on 26 March 2022; the data were recorded hourly. Because there were missing values, we selected 22,421 sets of data. We applied ten-fold cross-validation (CV) to obtain the prediction results and validate the process. In order to evaluate the model, the data were arranged chronologically, and the last 30% was taken as a hold-out test set. The remaining 70% of the data were used to train the model, and a 10-fold CV was applied. The averages of the CV evaluations provided evidence of the model's effectiveness on the test dataset. Using average evaluations, we tuned the model's parameters. Regarding the experimental software and tools, we adopted Weka 3.8.2 for clustering, logistic regression (LR), and MLPNN, as well as Pycharm 2017.1 for SVR and SVC. For GP, we used GPTIPS, an extension pack of MATLAB. Our MATLAB version was R2017b.

#### 5.1. Site Clustering

To predict the PM$_{2.5}$ concentrations more accurately, we first used the shortest distance clustering method to cluster the correlation coefficients of PM$_{2.5}$ concentrations at each site. The distance between the clusters was set at 0.3, and the 11 monitoring sites were divided into eight clusters, as shown in Table 5.

**Table 5.** Clusters with different intervals.

| Cluster | 1 | 2 | 3 | 4 | 5 | 6 | 7 | 8 |
|---|---|---|---|---|---|---|---|---|
| Site number (s) | 1 | 2 | 3 | 4, 11 | 5, 9 | 6, 8 | 7 | 10 |

### 5.2. Regression

The performances of LR, SVR, MLPNN, and GP were compared by predicting the $PM_{2.5}$ index at different time intervals of 1 h, 6 h, and 24 h. In the 1 h interval group, we considered the influence of the backtracking point. We used the root-mean-square error (RMSE) to evaluate the pros and cons of the model, as demonstrated by the RMSE $= \sqrt{\frac{\sum(predictPM - actualPM)^2}{n}}$. A lower RMSE indicated that the predicted value was closer to the actual value and that the model was more accurate.

The purpose of the parameter settings of all the predicting methods was that the RMSE was the lowest, while the time and space complexities were in an acceptable range. The parameters are shown in Tables 6 and 7. The parameters that were not listed would not likely affect the results; thus, we used the default values.

**Table 6.** The parameter settings of the proposed SVR.

| Time Intervals (h) | SVM Type | Degree | Gamma |
|:---:|:---:|:---:|:---:|
| 1 | epsilon-SVR | 3 | 0.1429 (without backtracking point) 0.125 (with backtracking point) |
| 6, 24 | epsilon-SVR | 3 | 0.1429 |

**Table 7.** The parameter settings of the proposed MLPNN.

| Time Intervals (h) | Decay | Learning Rate | Training Time (s) |
|:---:|:---:|:---:|:---:|
| 1, 6, 24 | TRUE | 0.2 | 1000 |

For the sake of brevity, we only included experimental results of 1 h prediction, as shown in Table 8. To decide whether to choose a backtracking point by using a correlation coefficient or distance, we experimented on both ways and compared their errors. In Table 8, "c" denotes using a correlation coefficient to obtain the backtracking point and "d" denotes using distance to obtain the backtracking point. As can be seen from the results, using a correlation coefficient to obtain the backtracking point was better than using distance. The RMSEs not marked with 'c' or 'd' mean that the backtracking points obtained using the correlation coefficient and distance were the same. The results of 1 h intervals indicate that without the backtracking point-based features, the approximate RMSEs of both the LR and the MLPNN were 11 to 27, the approximate RMSE of the SVR was 30 to 60, and the approximate RMSE of GP was 10 to 26, which was lower than other methods. When the related site concept was added, almost every method was reduced, but the reduction was not obvious. For the 6 h interval, the RMSEs of LR, MLPNN, SVR, and GP were separately approximately 28 to 51, 24 to 44, 33 to 60, and 23 to 44, respectively. The RMSEs of the four methods in the 24 h interval were 27 to 50 (LR), 26 to 44 (MLPNN), 33 to 60 (SVR), and 25 to 43 (GP). It can be inferred from the results that GP performed better than all the other methods with all the different time intervals; however, the reductions caused by related site features were observed but not evident. Furthermore, when the time interval increased, the RMSEs of all the methods increased. The predicted RMSEs of some sites were obviously higher than others in each time interval and each method, which was caused by data fluctuations at those sites.

**Table 8.** RMSEs at the 1 h interval on each site.

| | Site | LR with BP | LR without BP | MLPNN with BP | MLPNN without BP | SVR with BP | SVR without BP | GP with BP | GP without BP |
|---|---|---|---|---|---|---|---|---|---|
| | | | | | **Root-Mean-Square Error** | | | | |
| Individual sites | 1 | 12.08 | 12.14 | 11.89 | 11.89 | 33.77 | 33.77 | 10.57 | 10.99 |
| | 2 | 27.79 | 27.88 | 26.89 | 26.94 | 60.53 | 60.53 | 25.77 | 26.28 |
| | 3 | 16.60 | 17.39 | 16.45 | 16.58 | 48.52 | 48.53 | 15.65 | 15.70 |
| | 4 | 16.52 | 16.64 | 16.28 | 16.29 | 41.03 | 41.03 | 15.48 | 15.56 |
| | 5 | 22.47 | 22.93 | 21.22 | 21.39 | 53.59 | 53.59 | 20.51 | 20.93 |
| | 6 | 15.68 (c), 15.68 (d) | 15.73 | 15.03 (c), 15.03 (d) | 15.34 | 41.99 (c), 41.99 (d) | 41.99 | 14.27 (c), 14.49 (d) | 14.69 |
| | 7 | 22.74 | 22.89 | 22.4372 | 22.44 | 55.2668 | 55.28 | 20.87 | 21.10 |
| | 8 | 14.76 (c), 14.86 (d) | 14.85 | 14.15 (c), 14.16 (d) | 14.23 | 41.84 (c), 41.84 (d) | 41.85 | 13.77 (c), 13.81 (d) | 14.14 |
| | 9 | 19.081 | 19.14 | 18.84 | 19.059 | 49.99 | 49.99 | 17.81 | 18.34 |
| | 10 | 18.61 (c), 18.61 (d) | 18.66 | 18.25 (c), 18.25 (d) | 18.28 | 48.15 (c), 48.15 (d) | 48.16 | 18.01 (c), 18.29 (d) | 18.13 |
| | 11 | 15.83 | 15.98 | 15.57 | 15.57 | 38.70 | 38.79 | 15.14 | 15.41 |
| Clusters | 1 | 12.07 | 12.14 | 11.89 | 11.89 | 33.77 | 33.77 | 10.57 | 10.99 |
| | 2 | 27.79 | 27.88 | 26.89 | 26.94 | 60.53 | 60.53 | 25.77 | 26.28 |
| | 3 | 16.59 | 17.39 | 16.45 | 16.58 | 48.52 | 48.53 | 15.65 | 15.70 |
| | 4, 11 | 16.35 | 16.49 | 15.7913 | 15.84 | 41.96 | 41.97 | 15.51 | 15.52 |
| | 5, 9 | 20.50 | 20.66 | 20.03 | 20.24 | 53.85 | 53.94 | 19.85 | 19.97 |
| | 6, 8 | 14.87 (c), 14.87 (d) | 15.09 | 14.39 (c), 14.39 (d) | 14.68 | 43.68 (c), 43.68 (d) | 43.6866 | 14.01 (c), 14.02 (d) | 14.38 |
| | 7 | 22.73 | 22.89 | 22.4372 | 22.44 | 55.26 | 55.2848 | 20.8766 | 21.10 |
| | 10 | 18.61 (c), 18.61 (d) | 18.66 | 18.25 (c), 18.25 (d) | 18.28 | 48.15 (c), 48.15 (d) | 48.1608 | 18.01 (c), 18.29 (d) | 18.13 |
| All sites | 1 to 11 | 19.23 (c), 19.23 (d) | 19.36 | 19.94 (c), 19.94 (d) | 19.90 | 49.70 (c), 49.70 (d) | 49.7097 | 18.89 (c), 19.00 (d) | 19.00 |

Notes: the term "LR with BP" indicates the results of a logistic regression with backtracking points. Other terms "with BP" or "without BP" are in the same situation. The symbol "(c)" means that a correlation coefficient is used to obtain the backtracking point; the symbol "(d)" means that distance is used to obtain the backtracking point.

### 5.3. Tendency Classification

Using nu-SVC, we predicted whether PM$_{2.5}$ concentrations rose or fell in an hour. The measurement was defined as Recall $= \frac{TP}{TP+FN}$, i.e., a kind of variation in the original index recall from a confusion matrix, calculated as the ratio between the number of positive samples correctly classified as positive and the total number of positive samples. In this scenario of predicting PM$_{2.5}$ concentrations, people care more about the rises than the falls. Therefore, in this paper, we looked more at the prediction accuracy when the concentration rises. Let $T$ represent that the PM$_{2.5}$ index rises or is equal and $F$ represents that it falls, $TP$ indicates that the prediction is $T$ and the actual answer is $T$, and $FN$ indicates that the prediction is $F$ and the actual answer is $F$. Consequently, it is evident the higher the recall, the more accurate the model's predictions when air pollution is a concern. We used recall to evaluate our model; the results are shown in Table 9.

Overall, the average recall was approximately 0.7915. It can be inferred that the prediction targeting the tendency of PM$_{2.5}$ concentrations had some effects. In particular, recalls were higher at the 1 h interval with a backtracking point than without a backtracking point, but this was not obvious. Moreover, recalls did not decrease significantly when the time interval increased.

**Table 9.** Recall in tendency classification.

| Site | | Recall | | | |
| | | 1 h Interval with BP | 1 h Interval without BP | 6 h Interval | 24 h Interval |
|---|---|---|---|---|---|
| Individual sites | 1 | 0.79 | 0.79 | 0.79 | 0.80 |
| | 2 | 0.79 | 0.79 | 0.72 | 0.76 |
| | 3 | 0.79 | 0.78 | 0.8 | 0.79 |
| | 4 | 0.78 | 0.77 | 0.79 | 0.78 |
| | 5 | 0.79 | 0.78 | 0.79 | 0.80 |
| | 6 | 0.79 (c) 0.79 (d) | 0.79 | 0.79 | 0.78 |
| | 7 | 0.80 | 0.79 | 0.79 | 0.79 |
| | 8 | 0.78 (c) 0.79 (d) | 0.77 | 0.8 | 0.79 |
| | 9 | 0.79 | 0.78 | 0.79 | 0.78 |
| | 10 | 0.79 (c) 0.79 (d) | 0.79 | 0.797 | 0.80 |
| | 11 | 0.8 | 0.79 | 0.79 | 0.79 |
| Clusters | 1 | 0.79 | 0.79 | 0.79 | 0.80 |
| | 2 | 0.79 | 0.79 | 0.79 | 0.76 |
| | 3 | 0.79 | 0.78 | 0.80 | 0.79 |
| | 4, 11 | 0.79 | 0.79 | 0.78 | 0.79 |
| | 5, 9 | 0.79 | 0.79 | 0.78 | 0.79 |
| | 6, 8 | 0.79 (c) 0.79 (d) | 0.78 | 0.78 | 0.78 |
| | 7 | 0.80 | 0.79 | 0.79 | 0.79 |
| | 10 | 0.79 (c) 0.79 (d) | 0.79 | 0.79 | 0.80 |
| All sites | 1 to 11 | 0.79 (c) 0.79 (d) | 0.77 | 0.77 | 0.78 |

The symbol "(c)" means that a correlation coefficient is used to obtain the backtracking point; the symbol "(d)" means that distance is used to obtain the backtracking point.

### 5.4. Feature Influence

We used GP to obtain a more intuitive model to analyze the effect of the factors on $PM_{2.5}$ concentration. Table 10 lists the most influential factors in each set of experiments in each time interval. The results of the 1 h interval indicate that the most influential factors were the $PM_{2.5}$ concentrations in the previous period, the backtracking point, and the wind speed. Because the time interval was short, $PM_{2.5}$ concentrations in the previous period and the backtracking point were relatively important for prediction. However, as the time interval increased, other factors began to dominate the prediction, especially wind speed.

**Table 10.** Factors influencing the $PM_{2.5}$ concentrations.

| Sites | | Factors | | | |
| | | 1 h Interval with BP | 1 h Interval without BP | 6 h Interval | 24 h Interval |
|---|---|---|---|---|---|
| Individual sites | 1 | BP | PM | T | WS |
| | 2 | WS | WS | PM | WS |
| | 3 | PM | WS | WS | WS |
| | 4 | BP | WS | PM | WS |
| | 5 | BP | PM | WS | WS |
| | 6 | BP (c) WS (d) | WS | WS | WS |
| | 7 | WS | PM | WS | DH |
| | 8 | BP (c) DH (d) | DH | WS | DH |
| | 9 | WS | PM | RH | WS |
| | 10 | PM (c) PM (d) | PM | PM | WS |
| | 11 | PM | PM | WS | WS |
| Clusters | 1 | BP | PM | T | WS |
| | 2 | WS | WS | PM | WS |
| | 3 | PM | WS | WS | WS |
| | 4, 11 | BP | PM | WS | WS |
| | 5, 9 | PM | PM | WS | WS |
| | 6, 8 | BP (c) WS (d) | DH | WS | DH |
| | 7 | WS | PM | WS | DH |
| | 10 | PM (c) PM (d) | PM | PM | WS |
| All sites | 1 to 11 | DH WS | DH | WS | WS |

Notes: BP, backtracking point; PM, the $PM_{2.5}$ concentrations in the previous period; T, temperature; WS, wind speed; DH, in the dark hemisphere; RH, relative humidity. The symbol "(c)" means that a correlation coefficient is used to obtain the backtracking point; the symbol "(d)" means that distance is used to obtain the backtracking point.

## 6. Conclusions

Based on our experiments, we can draw many conclusions. First, GP dominated LR, MLPNN, and SVR in predicting $PM_{2.5}$ concentrations at time intervals of 1 h, 6 h, and 24 h, and the root-mean-square errors of GP were 10 to 26 (1 h), 23 to 44 (6 h), and 25 to 43 (24 h). Second, the new spatiotemporal feature produced a slight reduction in all the methods, with values ranging between 0.0001 and 0.01. We think the possible reasons are one or more of the following: (a) the wind direction and wind speed are constantly changing, which causes some inaccuracy in the choice of the backtracking point; (b) adapting the backtracking point concept to each method creates some differences; and (c) the concept of the backtracking point is itself incorrect or has some other insufficiency. Third, the $PM_{2.5}$ concentration tendency prediction based on SVC was relatively satisfactory with an average recall of 0.7915. Finally, GP provides an intuitive model for analyzing the factors which affect predictions. We found that when the time interval was relatively short, the most influential features were the $PM_{2.5}$ concentrations in the previous period and the backtracking points. Nevertheless, as the time interval increased, other factors, such as wind speed and the location of the predicted site, had a more significant influence on the prediction.

Our contributions in this study are as follows: (a) we innovatively applied GP to predict the $PM_{2.5}$ concentrations and verified the advantage of that method over LR, MLPNN, and SVR; (b) we innovatively proposed the backtracking point concept based on wind direction, wind speed, the correlation coefficient, and spatial geographic information; (c) we used SVC to predict the tendency of $PM_{2.5}$ concentrations and achieved some progress; and (d) we used GP to identify influential factors for $PM_{2.5}$ concentration prediction.

**Author Contributions:** Conceptualization, H.J.; methodology, X.W.; validation, C.S. and X.W.; formal analysis, X.W.; investigation, C.S.; resources, H.J.; writing—original draft preparation, X.W.; writing—review and editing, H.J.; visualization, X.W.; supervision, H.J.; project administration, C.S.; funding acquisition, H.J. All authors have read and agreed to the published version of the manuscript.

**Funding:** This research was funded by the National Natural Science Foundation of China (grant number 72071203).

**Institutional Review Board Statement:** Not applicable.

**Informed Consent Statement:** Not applicable.

**Data Availability Statement:** Not applicable.

**Acknowledgments:** This work was supported in part by the National Natural Science Foundation of China (Grant No. 72071203), and the Beijing Social Science Foundation (No. 20XCA001).

**Conflicts of Interest:** The authors declare no conflict of interest. The funders had no role in the design of the study; in the collection, analyses, or interpretation of data; in the writing of the manuscript; or in the decision to publish the results.

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
