# Peer review of "Predicting PM2.5 in the Northeast China Heavy Industrial Zone: A Semi-Supervised Learning with Spatiotemporal Features"

_atmosphere, doi:10.3390/atmos13111744_

Round 1

Reviewer 1 Report

The main contribution of the manuscript is the illustration of semi-supervised learning combined with genetic programming and support vector classification to predict the severity of multiple air pollutants. In the empirical study section of the manuscript, it has been shown that the proposed model outperforms multiple linear regression, multi-layer perceptron neural network, support vector regression, etc. The empirical study also shows that adding the spatial and temporal predictors improve the accuracy of predictions. I do feel that the topic and the architecture used for predicting the air pollutants in this manuscript is interesting, timely, and appropriate for Atmosphere.

I would suggest the revision of the manuscript to address the following issues:

1. The exposition needs significant improvement. The manuscript should expand on the technical details of the framework. I find that technical descriptions on Figure three, genetic programming, nu-SVC, etc., are too brief and not self-contained. Similarly, expanding on semi-supervised literature applied to similar problems would be useful. Also, be brief in explaining trivial topics like linear regression and cross validation. Almost all of the readers in this field would know about LR and CV.

2.The style of writing should be improved. At several places, confusing compound sentences were found in the manuscript.

3. Most of the references are before 2010. Is there any latest research in this field? The authors should add the latest research.

Some minor issues: 

Page 1:

Rewrite the following sentence/passage: “China suffers air pollution of particulate matter PM2.5, especially for a city like Shenyang locates in Northeastern China, the area of traditional heavy industries”. Here It seems that several prepositions use may be incorrect here.

“They are air-monitoring and meteorological data, and their spatiotemporal information, such as correlations between different locations (the spatiotemporal effects of PM2.5 emission and diffusion).” This sentence may be unclear or hard to follow. Consider rephrasing.

Page 2:

we present briefly a literature review in machine learning models.This sentence should be “we briefly present a literature review in machine learning models.

Page 3:

Rewrite the following sentence/passage:  prevent harms to fisheries and provide suggestion”. It seems that harms may not agree in number with other words in this phrase.

Page 5:

Rewrite the following sentence/passage:are used for generation of new offspring populations”, Here needs an article usage.

Page 7/9:

The subsections 5.2 and 5.3 have a same title “Regression”. Correct them.

Page 10:

' during the last period ' is a better term to use instead of ' in the previous period '.

Page 11:

This sentence is too hard to read. “We found that when the time interval was relatively short, the most influential features were the PM2.5 concentrations in the previous period and the backtracking points, but as the time interval increased, other factors such as wind speed and whether the predicted site was in the dark hemisphere had more of an effect on predictions.  Consider removing any unnecessary words OR splitting it into two sentences

Other issues:

1. Figure 1 is not clear.

2. The border of Figure 2 is inconsistent.

3. Figure 3 should add yes and no.

4. The words in Figure 8, Table 9 and Table 10 should not be separated from each other.

5. Some data in the table should be added with units.

Author Response

Response Sheet

Manuscript ID: atmosphere-1970352

Type: Article

Title: Predicting PM2.5 in Northeast China Heavy Industrial Zone: A semi-supervised learning with spatiotemporal features

Authors: Hongxun Jiang , Xiaotong Wang , Caihong Sun *

Reviewer 1 comments:

The main contribution of the manuscript is the illustration of semi-supervised learning combined with genetic programming and support vector classification to predict the severity of multiple air pollutants. In the empirical study section of the manuscript, it has been shown that the proposed model outperforms multiple linear regression, multi-layer perceptron neural network, support vector regression, etc. The empirical study also shows that adding the spatial and temporal predictors improve the accuracy of predictions. I do feel that the topic and the architecture used for predicting the air pollutants in this manuscript is interesting, timely, and appropriate for Atmosphere.

I would suggest the revision of the manuscript to address the following issues:

  1. The exposition needs significant improvement. The manuscript should expand on the technical details of the framework. I find that technical descriptions in Figure three, genetic programming, nu-SVC, etc., are too brief and not self-contained. Similarly, expanding on semi-supervised literature applied to similar problems would be useful. Also, be brief in explaining trivial topics like linear regression and cross validation. Almost all of the readers in this field would know about LR and CV.

Thank you for your suggestions. Now we expand the descriptions of the proposed framework with more technical details. That includes but not limited to the passages describing figure 3, GP, nu-SUV, etc. Meanwhile, we have shortened the passages on trivial topics like linear regression and cross-validation.

2. The style of writing should be improved. At several places, confusing compound sentences were found in the manuscript.

To make the paper as accurate as possible, it has been completely rewritten. We corrected several typos and grammar problems. There were some original sentences that were confusing. For example, "Most critically for the proposed models are that PM2.5 concentration from both the site predicted and its related sites nearby called backtracking points.” We rewrite it as “Historical PM2.5 concentrations from the target site and nearby sites as backtracking points are critical aspects of the proposed models.”

  1. Most of the references are before 2010. Is there any latest research in this field? The authors should add the latest research.

We add several latest references as follows,

  1. Liu, W., et al., Meteorological pattern analysis assisted daily PM2. 5 grades prediction using SVM optimized by PSO algorithm. Atmospheric Pollution Research, 2019. 10(5): p. 1482-1491.
  2. Yang, H., Z. Liu, and G. Li, A new hybrid optimization prediction model for PM2. 5 concentration considering other air pollutants and meteorological conditions. Chemosphere, 2022. 307: p. 135798.

Some minor issues: 

Page 1:

Rewrite the following sentence/passage: “China suffers air pollution of particulate matter PM2.5, especially for a city like Shenyang locates in Northeastern China, the area of traditional heavy industries”. Here It seems that several prepositions use may be incorrect here.

“They are air-monitoring and meteorological data, and their spatiotemporal information, such as correlations between different locations (the spatiotemporal effects of PM2.5 emission and diffusion).” This sentence may be unclear or hard to follow. Consider rephrasing.

We rewrite the sentence “China suffers air pollution of particulate matter PM2.5, especially for a city like Shenyang located in Northeastern China, the area of traditional heavy industries”, which is replaced by a new one “China suffers from air pollution of particulate matter PM2.5, especially in a city like Shenyang in Northeastern China, the area of traditional heavy industries.”

The sentence “They are air-monitoring and meteorological data, and their spatiotemporal information, such as correlations between different locations (the spatiotemporal effects of PM2.5 emission and diffusion)” is rewritten as “There are three types of data; air monitoring, meteorological data, and spatiotemporal information, such as the spatiotemporal effects of PM2.5 emissions and diffusion across different geographical regions.”

Page 2:

“we present briefly a literature review in machine learning models.” This sentence should be “we briefly present a literature review in machine learning models.”

Yes. We accept your suggestion and shorten this sentence as “Section 2 briefly presents a literature review of machine learning models.”.

Page 3:

Rewrite the following sentence/passage:  “prevent harms to fisheries and provide suggestion”. It seems that harms may not agree in number with other words in this phrase.

Yes. It was a grammar error and now we have corrected it. The new one is “prevent harms to fisheries and provide suggestions”

Page 5:

Rewrite the following sentence/passage: “are used for generation of new offspring populations”, Here needs an article usage.

We correct the sentence as “are used for the generation of new offspring populations.”

Page 7/9:

The subsections 5.2 and 5.3 have a same title “Regression”. Correct them.

That was a typo. The title of 5.3 should be “Tendency classification” and now we have corrected it.

Page 10:

' during the last period ' is a better term to use instead of ' in the previous period '.

Thank you for your suggestion. We replace it.

Page 11:

This sentence is too hard to read. “We found that when the time interval was relatively short, the most influential features were the PM2.5 concentrations in the previous period and the backtracking points, but as the time interval increased, other factors such as wind speed and whether the predicted site was in the dark hemisphere had more of an effect on predictions.”  Consider removing any unnecessary words OR splitting it into two sentences

 The sentence was too long to read. So we divide it into two sentences and rewrite to make them more readable, as “We found that when the time interval was relatively short, the most influential features were the PM2.5 concentrations in the previous period and the backtracking points. Nevertheless, as the time interval increased, other factors, such as wind speed and the location of the predicted site, had a more significant influence on the prediction.”

Other issues:

  1. Figure 1 is not clear.

We replace it with a better one in quality.

  1. The border of Figure 2 is inconsistent.

OK, It was. We correct it.

  1. Figure 3 should add yes and no.

Yes, we redrew a new figure.

  1. The words in Figure 8, Table 9 and Table 10 should not be separated from each other.

Yes, we adjust them to fit the limited space of the format.

  1. Some data in the table should be added with units.

We add the unit in table 1. The second column has a new caption, “Mean value of PM2.5 (μg/m3)”

Reviewer 2 Report

Please see attached file for comments. 

Author Response

Response Sheet

Manuscript ID: atmosphere-1970352

Type: Article

Title: Predicting PM2.5 in Northeast China Heavy Industrial Zone: A semi-supervised learning with spatiotemporal features

Authors: Hongxun Jiang , Xiaotong Wang , Caihong Sun *

Reviewer 2 comments:

This work presents the results of a proposed semi-supervised learning model for PM2.5  predictions in terms of mass concentration. The authors used data from 11 monitoring stations in Shenyang, China comprising of air quality monitoring and meteorological data, as well as their spatiotemporal information. The proposed model is consisting of a genetic programming model to predict PM2.5 mass concentration and a support vector classification model to predict the trends.

This manuscript presented interesting results and is generally well-written. However, often the visualizations (i.e. tables and figures) are often poorly constructed. There are important clarifications that need to be addressed and necessary improvements, such as more elaborate discussions and accurate take-aways in line with the results presented in this manuscript.

General comments:

Line 16: Do you mean MLR instead of MLP?

Yes, it was a typo. Now we correct it.

Line 36 to 38: Please provide reference for this line.

There are several works delivering the evidence about the relationship between meteorological data and PM2.5 concentrations. We two provide references as followed,

  1. Liu, W., et al., Meteorological pattern analysis assisted daily PM2. 5 grades prediction using SVM optimized by PSO algorithm. Atmospheric Pollution Research, 2019. 10(5): p. 1482-1491.
  2. Yang, H., Z. Liu, and G. Li, A new hybrid optimization prediction model for PM2. 5 concentration considering other air pollutants and meteorological conditions. Chemosphere, 2022. 307: p. 135798.

Line 38 to 39: Quite a confusing sentence, please rewrite. Are you defining backtracking points? Please elaborate.

The original sentence was confusing: "Most critically for the proposed models are that PM2.5 concentration from both the site predicted and its related sites nearby called backtracking points.” We rewrite it as “Historical PM2.5 concentrations from the target site and nearby sites as backtracking points are critical aspects of the proposed models.”

Line 52: Please add the word be between can and trace.

Yes, here we missed a word “be”. And we rewrite it as “The first attempt to predict air quality can be traced back to Miller et al. (1976), who used two different models based on Gaussian dispersion equations to predict carbon monoxide concentrations.”

Figure 2: The image quality needs to be greatly improved. If the goal of Figure 2 is to show the correlations of these variables (as discussed in Lines 127 to 133), then perhaps a correlation matrix or a scatterplot matrix would provide better visualization.

Yes, we replace the figures by better ones in quality.

Line 140 to 142: Please clarify what backtracking points are and how are they estimated. Or mention that it will be discussed further in section 4.

Yes, we clarify them in section 4. See there for more detailed information.

Section 3: I would assume that this section discusses about feature importance as well. Did you apply a feature selection step? Please discuss why or why not.

Actually, we do not have a step for feature selection. The entries in data we have so far are not too rich in diversity. If some features are indeed not beneficial for the prediction of PM2.5 concentrations, the subsequent classifiers could automatically exclude them as well.

Table 1: Please mention the unit of the concentration in the table caption.

Yes, good idea. We add it.

Table 8: Please consider using a heatmap with annotations or a categorical plot to visualize these results. Currently, it is quite difficult to catch any take-aways from the table.

Acturally we have tried to apply a heatmap with annotations to visualize the results, but it makes people more confused since there are too many entries in a table resulting in a clutter of different colors.

Line 282 to 286: Please define PM2.5 index. Please also clarify if the variables T and F are binary variables. This paragraph is very important and needs to be improved.

Here the PM2.5 index means the PM2.5 concentration, so we rewrite the word “index” replaced by “concentrations”. The original descriptons of variables T and F were confused, so we add more information about this measurement. “The measurement is defined as Recall=TP/(TP+FN). It is a kind of variation of the original index Recall from a confusion matrix, which is is calculated as the ratio between the number of Positive samples correctly classified as Positive to the total number of Positive samples. In this scenario of predicting PM2.5 concentrations, people care more the rises than the falls. Therefore, in this paper, we look more at the prediction accuracy when the concentration rises. Let T represents that the PM2.5 index rises or is equal and F represents that it falls, TP indicates that the prediction is T and the actual answer is T, and FN indicates that the prediction is F and the actual answer is F. That It is not difficult to conclude that the higher the recall, the more accurate the prediction results of the model when people really need worry about air pollution. We used recall to evaluate our model; the results are shown in Table 9.”

Line 302 to 304: It is not surprising that other factors begin to dominate as time resolution was increased. There will be stronger variations in the meteorological variables as you increase the time resolution due to diurnal variability. This is where a feature selection step or feature importance step should come useful.

Absolutely agree with you opinion that diurnal variability will result in more substantial variations in the meteorological variables as the time resolution increases. Since the features are not particularly large in this study, we still leave the filtering and sifting of the features to the classifier.

Line 330 to 331: some progress is ambiguous and needs to be specified.

It was ambiguous, and we rewrite it as “Nevertheless, as the time interval increased, other factors, such as wind speed and the location of the predicted site, had a greater influence on the prediction.”.

Other comments:

Did you consider any resampling method to allow estimation of accuracy to the predictions?

Yes, we have actually applied ten-fold cross-validation to obtain the prediction results and validate the process. The description of this part of the work was omitted from the previous version of the paper. This time we add it into the beginning of Section 5, as “In order to evaluate the model, the data were arranged chronologically, and the last 30% was taken as a hold-out test set. The remaining 70% of the data was used to train the model, and a 10-fold CV was applied. The averages of the CV evaluations provided evidence of the model's effectiveness on the test dataset. Using average evaluations, we tuned the model's parameters.”

The methodology of the other models compared to the GP model is not clear. All methodologies of the models compared should be reported for better comparison.

We should do that. So in this one version, we have added many technical details of the comparison model.

Round 2

Reviewer 1 Report

The author has revised it according to my opinion.